# N-Acetyl cysteine exhibits antimicrobial and anti-virulence activity against *Salmonella enterica*

**Selwan Hamed**[1]*, **Mohamed Emara**[1], **Payman Tohidifar**[2], **Christopher V. Rao**[2]

**1** Department of Microbiology and Immunology, Faculty of Pharmacy, Helwan University− Ain Helwan, Helwan, Cairo, Egypt, **2** Department of Chemical and Biomolecular Engineering, University of Illinois at Urbana-Champaign, Urbana, Illinois, United States of America

* Silwan_mustafa@pharm.helwan.edu.eg

**Data Availability Statement:** All relevant data are within the manuscript and its Supporting information files. DNA sequencing are available under the following link https://www.ncbi.nlm.nih.gov/sra/PRJNA878837.

## Abstract

*Salmonella enterica* is a common foodborne pathogen that causes intestinal illness varying from mild gastroenteritis to life-threatening systemic infections. The frequency of outbreaks due to multidrug-resistant *Salmonella* has been increased in the past few years with increasing numbers of annual deaths. Therefore, new strategies to control the spread of antimicrobial resistance are required. In this work, we found that N-acetyl cysteine (NAC) inhibits *S. enterica* at MIC of 3 mg ml$^{-1}$ and synergistically activates the bactericidal activities of common antibiotics from three-fold for ampicillin and apramycin up to 1000-fold for gentamycin. In addition, NAC inhibits the expression of virulence genes at sub-inhibitory concentrations in a dose-dependent manner. The whole-genome sequencing revealed that continuous exposure of *S. enterica* to NAC leads to the development of resistance; these resistant strains are attenuated for virulence. These results suggest that NAC may be a promising adjuvant to antibiotics for treating *S. enterica* in combination with other antibiotics.

## Introduction

*Salmonella enterica (S. enterica)* is a bacterial pathogen that is capable of infecting a wide range of hosts [1]. While infections could easily be treated with antibiotics in the past, *S. enterica* is rapidly developing resistance to many antibiotics [2]. Ideally, these antibiotics-resistant infections could be treated with new antibiotics. However, developing new antibiotics is a challenging task, taking years of research and millions of dollars to develop [3]. In the meantime, these antibiotic-resistant bacteria continue to develop more resistance, leading to more death due to systemic infections [4].

Rather than trying to develop entirely new antibiotics, an alternate strategy is to repurpose existing drugs to treat bacterial infections [5]. In this work, we explored N-acetyl cysteine (NAC) as a potential antimicrobial agent against *S. enterica*. NAC is derived from the amino acid L-cysteine and is widely available as an over-the-counter medication [6, 7]. Medical use of NAC began in the 1960s as an antidote for acetaminophen poisoning [8]. Since then, its clinical and medical applications have expanded as an antioxidant for the treatment of diabetes,

**Funding:** The author(s) received no specific funding for this work.

**Competing interests:** All authors declare that there is no conflict of interest.

neurological conditions, and liver diseases [9]. In addition, NAC is used as a mucolytic agent and adjuvant to antibiotics therapy in the treatment of upper respiratory tract infections [10] by reducing the viscosity of tracheobronchial mucous owing to its ability to break disulfide linkages of the mucin [11].

Some have suggested that NAC may be an effective antibiotic for treating bacterial infections [12, 13]. Because of its high safety and wide therapeutic index, it can be safely tolerated at doses of 500–1200 mg/day, and 4000 mg/day when treating some neurological disorders [14]. In this work, we tested the effectiveness of NAC as an antibiotic for *S. enterica*, alone, and in combination with other antibiotics. In addition, we tested whether *S. enterica* is capable of evolving resistance against NAC. While evolved strains (ES) did exhibit increased resistance to NAC, we observed that the ES also exhibited reduced expression of their virulence genes. These results suggest that NAC may be a promising adjuvant for treating antibiotic-resistant *Salmonella* infections.

## Results

### NAC synergistically activates the antimicrobial effect of antibiotics

N-acetyl cysteine (NAC) is a synthetic precursor for glutathione that is used as an antioxidant and adjuvant to antibiotics for treating upper respiratory tract infections [10]. Recent studies have demonstrated the antimicrobial activity of NAC against bacteria such as *Escherichia coli* and *Pseudomonas aeruginosa* [15]. In this work, we investigated NAC's antimicrobial activity against *S. enterica*.

We first used the broth microdilution method to ascertain that the minimum inhibitory concentration (MIC) of NAC is 3 mg ml$^{-1}$. Next, we explored whether NAC could enhance the activity of other common antibiotics: ampicillin, apramycin, chloramphenicol, gentamycin, kanamycin, and streptomycin. This was determined by estimating the fractional inhibitory concentration (FIC) using the checkerboard technique [15, 16]. When the FIC index is less than 0.5, then the two drugs have a synergistic effect [17]. All NAC and antibiotics combinations showed FIC values less than 0.5, which indicates that NAC has a synergistic effect when combined with other common antibiotics (Table 1).

In particular, the addition of NAC reduced the MIC for these common antibiotics. The degree varied depending on the antibiotic: from three and a half folds for ampicillin and apramycin and up to a thousand-fold for gentamycin. Collectively, these results suggest that NAC can be used to increase to effectiveness of some antibiotics against *S. enterica* infections.

**Table 1. Minimum inhibitor concentrations (MICs) for each tested compound alone and fractional inhibitor concentrations (FICs) when tested with N-acetyl cysteine.**

| | | NAC | CM | Gen | Amp | Strep | KM | Apr |
|---|---|---|---|---|---|---|---|---|
| **MIC (µg/ml)** | | 3000* | 3 * | 0.6 * | 25** | 10* | 5 ** | 25** |
| **Concentration in the Mix (mg/ml)** | AB | | 0.085** | 0.000681* | 7** | 0.285* | 0.127* | 7** |
| | NAC | | 1* | 1** | 0.5* | 1.5* | 1* | 0.5** |
| **ΣFIC** | | | 0.361** | 0.334* | 0.42* | 0.358* | 0.358* | 0.42** |

**NAC:** N-acetyl cysteine, **MIC:** Minimum inhibitory concentration, **CM:** Chloramphenicol, **Gen:** Gentamycin, **Amp:** Ampicillin, **Strep:** Streptomycin, **KM:** Kanamycin, **Apr:** Apramycin, **ΣFIC:** fractional inhibitor concentrations. The presented data represents mean of three replicates.

\* Means p-value $\leq$ 0.05,

\*\* means p-value $\leq$ 0.001.

## NAC represses virulence genes expression

We next investigated the ability of NAC to suppress virulence in *S. enterica*. NAC can treat otitis media, a biofilm-forming infection caused by *Streptococcus pneumoniae* and *Haemophilus influenza* [18–20]. In addition, NAC significantly improves the healing of *Pseudomonas aeruginosa* contaminated wounds by inhibiting biofilm development [21]. Since NAC can inhibit bacterial virulence, we investigated whether NAC, at subinhibitory concentrations, can downregulate the expression of virulence genes in *S. enterica*. We focused on the expression of HilA, which is the master regulator of invasion genes expression encoded within *Salmonella* pathogenicity island 1 [22]; these genes are required for the invasion of intestinal epithelial cells. To measure the expression of HilA, we fused the green fluorescent protein (GFP) to the *hilA* promoter and then measured fluorescence using flow cytometry. In addition, we also tested whether NAC could inhibit motility gene expression in *S. enterica*. Motility is also a virulence factor in *S. enteric* [23–26]. Here, we focused on measuring the expression of the flagellin (FliC) again using flow cytometry by fusion of the fluorescent protein Venus to the *fliC* promoter [26–28].

We found that NAC, at a subinhibitory concentration of 2.5 mg ml$^{-1}$, reduced the expression from both the *hilA* and *fliC* promoters (Fig 1 and S1 Fig). The degree of repression was dose-dependent: as the concentration increased, promoter activity decreased. The mechanism is unknown. Nevertheless, these data demonstrate that NAC not only kills *S. enterica* but also potentially reduces its virulence.

## Evolution of resistance against NAC

Bacteria are capable of developing resistance to most antibiotics. Therefore, we tested whether *S. enterica* could develop resistance to NAC during prolonged exposure to sub-inhibitory concentrations of 1.5 mg ml$^{-1}$. In these experiments, we grew *S. enterica* for 30 days in either Luria-Bertani broth or M9 minimal medium containing glucose and were sub-cultured daily into fresh medium. As shown in Fig 2, *S. enterica* was able to tolerate the presence of NAC following prolonged exposure. In particular, growth was less inhibited in the ES than in the wild-type control. Surprisingly, the ES showed lower MIC of 2–2.2 mg ml$^{-1}$ for NAC compared to the wild type (Table 2).

## NAC-resistant strains are attenuated

We found that NAC inhibits the expression of the virulence and motility genes in *S. enterica* at a sub-inhibitory concentration (Fig 1). An open question was whether the NAC-resistant strain also exhibited reduced expression of the virulence genes. To answer this question, we measured the expression of *hilA* and *fliC* promoters in the NAC-resistant strains using flow cytometry. We found that the expression of *hilA* and *fliC* was significantly (p< 0.05) reduced 4 and 6-fold, respectively, in the NAC-resistant strain as compared to wild-type (unevolved) *S. enterica* (Fig 3 and S2 Fig), that's also confirmed by motility plate assay shown in (Fig 4).

We used whole genome sequencing (WGS) technique to get insight and understand the underlying mechanism for virulence attenuation in ES. In this experiment, the whole DNA was extracted from wild type *S. enterica*, three biological replicates evolved in LB medium, and another three colonies evolved in M9 medium. The ES showed common modifying single nucleotide polymorphisms (SNPs) mostly in Salmonella NinG encoded on Lambda phage, Gifsy 2 prophage and its encoded genes as illustrated in (S1 Table) while some ES showed individual mutations in invasion genes SseI (SPI-2 type III secretion system effector) and tail fiber assembly protein.

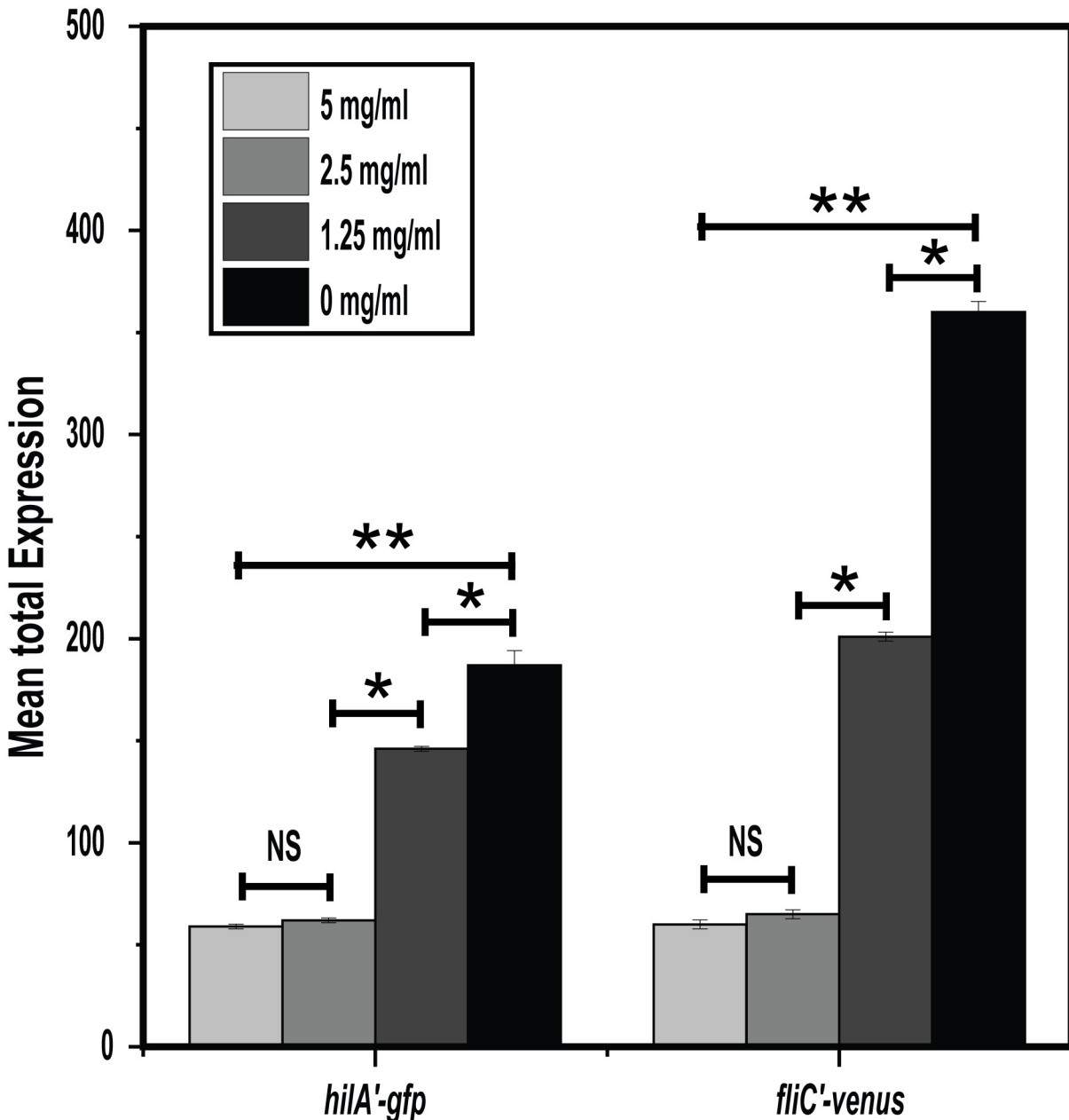

**Fig 1. NAC suppresses the expression of *Salmonella* virulence gene.** *hilA* and *fliC* promoters' activities were determined at varying concentrations of NAC using single-copy transcriptional fusions to the fluorescent proteins GFP and Venus, respectively. Mean total expression of promoters were significantly inhibited at high concentrations of NAC in a dose-dependent manner [NS: not significant, * p-value < 0.05, and ** p-value< 0.001].

## Discussion

*S. enterica* infections are common and vary from gastroenteritis to a more serious systemic infection. Most people get infected with *S. enterica* by eating contaminated meat, poultry, eggs, fruits, and vegetables [29, 30]. Since 2000, several outbreaks of multidrug-resistant *Salmonella* infections have been reported in both the United States and Europe [31–33]. Thus, there is an

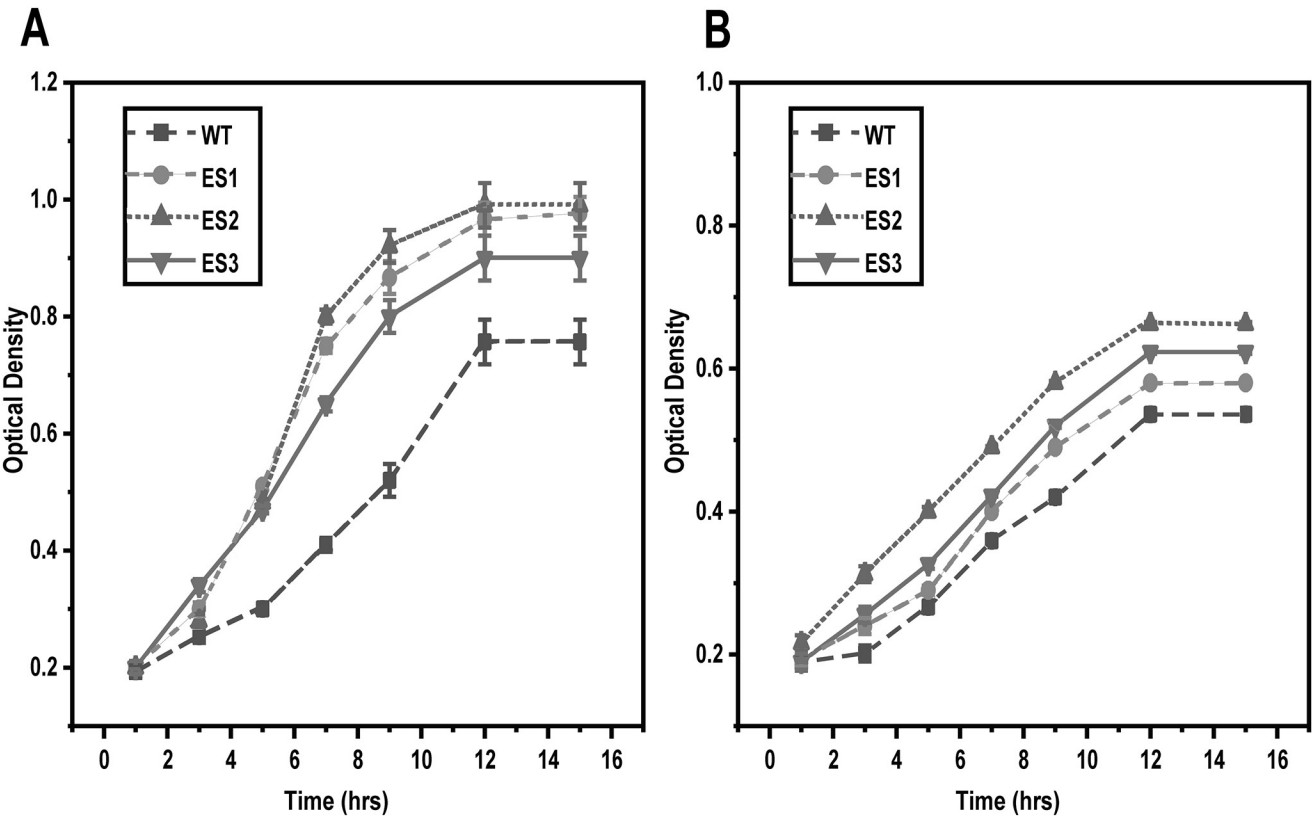

**Fig 2. *Salmonella* develops resistance to NAC.** Both NAC evolved strains and wild type control were grown in either (a) LB medium or (b) M9 medium supplemented with (1.5 mg ml $^{-1}$). The wild type of control showed inhibited growth as compared to the evolved strains (ES1-3) that tolerate the presence of NAC.

urgent need to combat bacterial resistance by employing alternatives to conventional antibiotics. Repurposing existing drugs is promising money and time-saving approach [5].

NAC is conventionally used as an antioxidant to replenish glutathione pools, which in turn scavenges free radicals and reactive oxygen species [34, 35]; thus, NAC not only detoxifies the body but also strengthens the immune system [9, 10]. In addition, it can also be used as an antibiotic. Indeed, growing experimental evidence has demonstrated the potential of using NAC as a safe antimicrobial agent, either as a monotherapy or combined with antibiotics [36–38].

These results motivated us to test the antimicrobial activity of NAC against *S. enterica*. We found that NAC not only inhibited the growth of S. *enterica* but also enhanced the activity of many common antibiotics. Our results are in line with previously published work that

**Table 2. Minimum inhibitor concentrations (MICs) for NAC for wild type versus evolved strains.** MIC results decrease when wild type S. enterica exposed to subinhibitory concentrations of NAC for 30 growth cycles.

|  | **WT** | **ES_LB** | **ES_M9** |
|---|---|---|---|
| **MIC for NAC (mg/ml)** | 3* | 2* | 2.2* |

**NAC:** N-acetyl cysteine, **MIC**: Minimum inhibitory concentration, **ES_LB:** evolved strains grown in LB medium,

**ES_M9:** Evolved strains grown in M9 medium. The presented data represents mean of three replicates.

* Means p-value $\leq$ 0.05

**Fig 3. NAC-resistant strains are attenuated.** *hilA* and *fliC* promotor activities were measured for the wild type (WT) and NAC evolved strains (ES) growing in LB medium using single-copy transcriptional fusions to GFP and Venus, respectively. The mean total expression of promoter activities in NAC-resistant strains was significantly inhibited (p-value < 0.05) as compared to wild-type control.

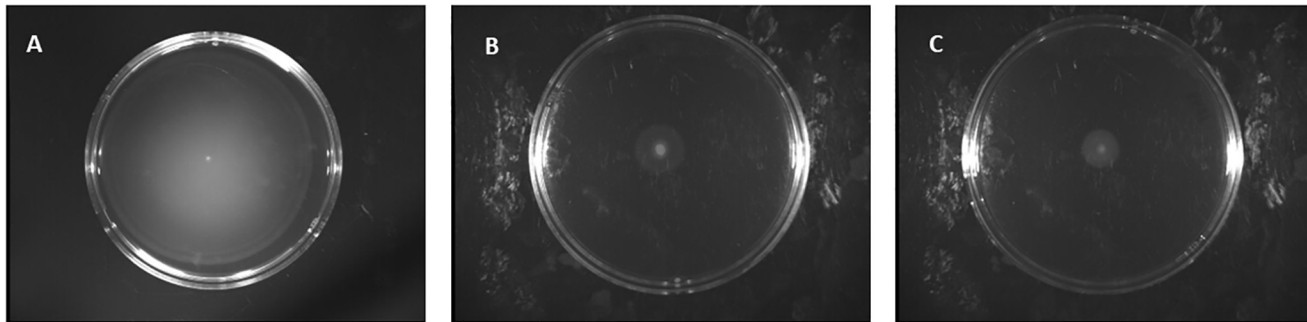

**Fig 4. Motility plate assay for NAC-resistant strains versus wild type.** Motility of control strain **A: wild type (WT)** showed a normal motility zone while NAC evolved strains (ES) **B:** growing in LB medium or **C: M9 medium** showed reduced motility.

suggested using NAC at concentrations ranging from 10–20 mM (1.6–3.2 mg ml$^{-1}$) can treat simple infections [39]. Of note, higher concentrations of NAC are required, approximately 50 mM (16 mg ml$^{-1}$), for treatment of mixed infections due to the inherent antibiotic resistance of the causative strains and the development of biofilm [40]. At daily doses of 1200–1800 mg/day, NAC was able to suppress *Pseudomonas aeruginosa* in cystic fibrosis patients by inhibiting quorum sensing, development of biofilm, and initiation of extracellular polysaccharides and bacterial matrix production [39, 41]. Likewise, the invasion and motility genes of *S. enterica* were inhibited at subinhibitory concentrations of NAC in a dose-dependent manner, while the continuous exposure to lower concentrations of NAC during bacterial growth leads to the attenuation of these virulence determinants in *Salmonella*.

The exact mechanism of NAC's antimicrobial activity is not known. It was previously believed that the thiol group interacts with bacterial disulfide bonds; however, recent studies proved that the substitution of the thiol group is not altering NAC effectiveness [21]. Meanwhile, both the acetyl and carboxyl groups are essential for the antimicrobial effect of NAC, where they affect protein folding and flagellar assembly [42]. NAC is a multi-targeting agent that can penetrate the bacteria, competitively inhibits cysteine utilization, potentially protect against oxidative stress, and suppresses protein synthesis [43].

The whole genome sequencing for ES compared to wild type *S. enterica* revealed common mutations in DUF proteins that were previously reported to inhibit the ability of *S. enterica* to induce systemic infection [44, 45]. Several studies revealed that prophage Gifsy 1 and 2 have a critical role in pathogenesis of *S. enterica* [46]; cured bacteria are not able to induce infection in mice [44]. That's aligned with our results as NAC evolved strains showed reduced expression of virulence genes related to motility and invasion, a possible mechanism that NAC interact at some level with bacteriophage.

Taken together, NAC is a promising candidate to combat antimicrobial resistance with a high safety profile. It also attenuates virulence in *S. enterica*. Further investigations are required to understand the potential NAC's mechanism of action in *S. enterica*.

## Materials and methods

### Bacterial strains, and general growth conditions

In all experiments, we used *Salmonella enterica* serovar Typhimurium ATCC 14028 (American Type Culture Collection) (*S. enterica)*. Growth experiments were performed using either Luria-Bertani (LB) medium or M9 minimal medium (supplemented with 0.2% glucose) at 37 °C, unless noted otherwise. To measure the expression of the invasion and motility gene, strains expressing either the *hilA* or *fliC* promoters respectively, were used as previously described [26, 28, 47]. All experiments were conducted in triplicate.

### Adaptive evolution

A single colony of wild-type *S. enterica* was streaked on the surface of an LB plate, then six colonies were picked, divided into two groups, and grown under shaking (200 rpm) in either LB or M9 media supplemented with 1.5 mg ml$^{-1}$ of NAC. Batch cultures were manually transferred to fresh media containing 1.5 mg ml$^{-1}$ of NAC each day. The process was repeated for 30 days.

### Assessment of antibacterial activity of N-acetyl cysteine (NAC) against S. enterica

The antimicrobial activity of NAC was determined using the broth microdilution technique [48, 49]. Briefly, we used starting concentrations of 50 mg ml$^{-1}$ for NAC that was then diluted

in a double fold manner across a 96-well plate. $1\times10^5$ CFU ml$^{-1}$ of *S. enterica* was added, cultures were then incubated under shaking (250 rpm) for 12–18 hours. The minimum inhibitory concentration (MIC) of NAC was determined by measuring the growth inhibition at an optical density (OD) of 600 nm.

## Synergy evaluation

We used the checkerboard dilution procedures to investigate the effect of NAC and tested antibiotics when present in combination [16, 17]. Briefly, in a 96- well plate, both NAC and the antibiotics were diluted along the x and y-axis. The combined effect and the fractional inhibitory concentration (FIC) index were obtained by measuring optical density at 600 nm using a Tecan microplate reader.

## Gene expression assay

Expression from the *hilA* and *fliC* promoters were measured as described previously using a BD Biosciences LSR II flow cytometer [26, 47]. Briefly, cells were grown in LB statically when measuring expression from the *hilA* promoter and under vigorous shaking (250 rpm) when measuring expression from the *fliC* promoter. The mean fluorescent intensities were obtained from 50,000–100,000 cells.

## DNA sequencing

A total of seven DNA samples were extracted from wild type and six colonies representing biological replicates evolved in either LB or M9 media. DNeasy blood and tissue kit (Qiagen-USA) was used to extract DNA as per manufacturer's instruction. DNA samples were sequenced at the Roy J. Carver Biotechnology Center, University of Illinois at Urbana-Champaign. Briefly, the shotgun genomic libraries were prepared with the Hyper Library construction kit from Kapa Biosystems (Roche). The libraries were pooled; quantitated by qPCR and sequenced on one SP lane for 101 cycles from one end of the fragments on a NovaSeq 6000. Fastq files were generated and demultiplexed with the bcl2fastq v2.20 Conversion Software (Illumina). Adaptors have been trimmed from the 3'-end of the reads. Sequence of adaptors used to make the libraries. The quality-scores line in fastq files use an ASCII offset of 33 known as Sanger scores.

FASTA files for the raw reads were submitted to NCBI under the following accession number PRJNA878837 SAMN30758791-97(https://www.ncbi.nlm.nih.gov/sra/PRJNA878837).

## Whole genome sequencing analysis

*Salmonella enterica subsp. enterica* serovar Typhimurium (ASM386401v1) was used as the reference genome. BWA/0.7.17-IGB-gcc-4.9.4 was used to align the data [50]. BCF tools/1.9-IGB-gcc-4.9.4 was used to call the variant and detect the single nucleotide polymorphisms (SNPs) [51]. SnpEff was installed and used to determine the effect of the SNPs [52].

## Statistical analysis

Statistical analyses were performed by using Origin pro 2019 (Origin Lab, Northampton, Massachusetts, USA). Student's t-tests were used for pairwise comparisons analysis. Significance was accepted when p values were less than 0.05 ($p < 0.05$). Data from three biological replicates are presented as means with standard deviations.

## Supporting information

**S1 Table. Results of Whole genomic DNA Sequencing.** All the modifying single nucleotide polymorphisms (SNPs) between wild type and evolved strains are listed with their potential effect on bacterial virulence.
(XLSX)

**S1 Fig. Expression of fluorescent protein in wild type *S. enterica* treated with different concentrations of NAC.** A: control, B: expression at 1.25 mg ml$^{-1}$ NAC, C: expression at 2.5 mg ml$^{-1}$NAC and D: expression at 5 mg ml$^{-1}$NAC.
(TIF)

**S2 Fig. Expression of fluorescent protein in wild type versus evolved strains *S. enterica* treated with Subinhibitory concentration of NAC.** A: Wild type, B: evolved strains.
(TIF)

## Author Contributions

**Conceptualization:** Selwan Hamed.

**Data curation:** Selwan Hamed, Mohamed Emara.

**Resources:** Christopher V. Rao.

**Software:** Payman Tohidifar.

**Supervision:** Christopher V. Rao.

**Writing – original draft:** Selwan Hamed.

**Writing – review & editing:** Mohamed Emara, Christopher V. Rao.

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
