## [Decision Letter · Decision Letter 0]

14 Oct 2024

PONE-D-24-31564N-acetyl cysteine exhibits antimicrobial and anti-virulence activity against Salmonella entericaPLOS ONE

Dear Dr. Aboelnaga,

Thank you for submitting your manuscript to PLOS ONE. After careful consideration, we feel that it has merit but does not fully meet PLOS ONE’s publication criteria as it currently stands. Therefore, we invite you to submit a revised version of the manuscript that addresses the points raised during the review process.

We look forward to receiving your revised manuscript.

Kind regards,

Anujith Kumar

Academic Editor

PLOS ONE

Journal Requirements: When submitting your revision, we need you to address these additional requirements. 1. Please ensure that your manuscript meets PLOS ONE's style requirements, including those for file naming. The PLOS ONE style templates can be found at https://journals.plos.org/plosone/s/file?id=wjVg/PLOSOne_formatting_sample_main_body.pdf and https://journals.plos.org/plosone/s/file?id=ba62/PLOSOne_formatting_sample_title_authors_affiliations.pdf 2. We suggest you thoroughly copyedit your manuscript for language usage, spelling, and grammar. If you do not know anyone who can help you do this, you may wish to consider employing a professional scientific editing service.  The American Journal Experts (AJE) (https://www.aje.com/) is one such service that has extensive experience helping authors meet PLOS guidelines and can provide language editing, translation, manuscript formatting, and figure formatting to ensure your manuscript meets our submission guidelines. Please note that having the manuscript copyedited by AJE or any other editing services does not guarantee selection for peer review or acceptance for publication.  Upon resubmission, please provide the following: The name of the colleague or the details of the professional service that edited your manuscript A copy of your manuscript showing your changes by either highlighting them or using track changes (uploaded as a *supporting information* file) A clean copy of the edited manuscript (uploaded as the new *manuscript* file)” 3. We note that the grant information you provided in the ‘Funding Information’ and ‘Financial Disclosure’ sections do not match.  When you resubmit, please ensure that you provide the correct grant numbers for the awards you received for your study in the ‘Funding Information’ section. 4. Thank you for stating the following financial disclosure: "This work was partially supported by a visiting fellowship from the Egyptian cultural and educational bureau – Washington, DC." Please state what role the funders took in the study.  If the funders had no role, please state: ""The funders had no role in study design, data collection and analysis, decision to publish, or preparation of the manuscript."" If this statement is not correct you must amend it as needed. Please include this amended Role of Funder statement in your cover letter; we will change the online submission form on your behalf. 5. Please ensure that you refer to Figure 4 in your text as, if accepted, production will need this reference to link the reader to the figure. 6. Please include a copy of Table 2 which you refer to in your text on page 5. 7. Please include captions for your Supporting Information files at the end of your manuscript, and update any in-text citations to match accordingly. Please see our Supporting Information guidelines for more information: http://journals.plos.org/plosone/s/supporting-information. 8. Please review your reference list to ensure that it is complete and correct. If you have cited papers that have been retracted, please include the rationale for doing so in the manuscript text, or remove these references and replace them with relevant current references. Any changes to the reference list should be mentioned in the rebuttal letter that accompanies your revised manuscript. If you need to cite a retracted article, indicate the article’s retracted status in the References list and also include a citation and full reference for the retraction notice.

**Additional Editor Comments:**

Dear Dr. Selwan,

Your manuscript has been evaluated by the Editors and qualified reviewer(s) and we are pleased to report that this work has been judged to be potentially suitable for publication in Plos One.

We request that you revise your manuscript in accordance with the comments below. Please note that the revised manuscript will require an additional evaluation by the Editors and that this request for revision is not a guarantee of final acceptance

Reviewers' comments:

Reviewer's Responses to Questions

**Comments to the Author**

1. Is the manuscript technically sound, and do the data support the conclusions?

Reviewer #1: Yes

Reviewer #2: Partly

2. Has the statistical analysis been performed appropriately and rigorously? 

Reviewer #1: Yes

Reviewer #2: Yes

3. Have the authors made all data underlying the findings in their manuscript fully available?

Reviewer #1: Yes

Reviewer #2: Yes

4. Is the manuscript presented in an intelligible fashion and written in standard English?

Reviewer #1: Yes

Reviewer #2: Yes

5. Review Comments to the Author

Reviewer #1: This study was designed to investigate the antimicrobial activity of NAC against S. enterica and to verify the potential of NAC as a promising antibiotic for treating S. enterica in combination with other antibiotics. The overall design of the paper is simple and reasonable. Comments: (1) The basis for selecting Salmonella ATCC 14028 needs to be supplemented. (2) The method for determining MIC needs to be supplemented, and the reason for not considering MBC as an indicator should be explained. (3) Figure 2 can be further optimized for display, such as using solid or hollow symbols, solid or dashed lines, etc. Is there no color version of Figure 4?

Reviewer #2: The present study tries to establish NAC as an adjuvant which can enhance the effect of antibiotics in treatment of Salmonella Enterica infection. The authors derive the minimal inhibitory concentration of various antibiotics in presence and absence of NAC and show that it can be a beneficial addition to the antibiotic pool. They also show that NAC treatment leads to reduced virulence of S. enterica. Though the study demonstrates the positive role of NAC in combatting S. enterica infection, there are major limitations in the study that need to be addressed.

Major queries:

1) The concentration of NAC used in the study is strikingly different from what is reported in one of the previous studies. Ondrej Chlumsky et.al., 2021 (Evaluation of the Antimicrobial Efficacy of N-Acetyl-l-Cysteine, Rhamnolipids, and Usnic Acid—Novel Approaches to Fight Food-Borne Pathogens) showed MIC of NAC to be 12500 μg/mL which is strikingly different from 3 mg/mL used in the present study. Could the authors justify the effect seen with a lower concentration of NAC?

2) The authors show that administration of NAC leads to decreased expression of virulence genes: hilA and fliC. It is important to understand what this would mean physiologically. Does the reduced virulence lead to less severe symptoms of S. enterica infection? Also, it is recommended that the authors show the expression of other important virulence genes such as invA (assists in the host invasion) and iroB (involved in the formation and transfer of enterobactin)

3) The results show that the enhanced strains show a lowered MIC for NAC. How would the authors explain this? And why do the virulence genes in NAC resistant strains respond to NAC treatment?

4) Changes in the WGS profile with NAC treatment would not give a substantial explanation to the changes in virulence. An explanation about the probable mechanism of NAC mediated effects on S. enterica need to be provided in the discussion section. Also, WGS data need to be validated.

Minor queries:

1) The authors use hilA and fliC promoters driven GFP expression as a readout of virulence. However, GFP expression is not shown in the manuscript. In addition, flow cytometry plots must be shown in the figure along with the quantification graph.

2) In the abstract, NAC is referred to as an antibiotic and as an adjuvant in another section. It would be better to maintain a uniform terminology. Also, referring to an antioxidant as an antibiotic might need to be reconsidered.

3) In the discussion section, line 148, it is stated that NAC increases oxidative stress. This misleading statement needs to be corrected.

6. PLOS authors have the option to publish the peer review history of their article (what does this mean?). If published, this will include your full peer review and any attached files.

Reviewer #1: **Yes: **Qingli Dong

Reviewer #2: No

---

## [Author Response · Author response to Decision Letter 0]

21 Oct 2024

5. Review Comments to the Author

Response to reviewers’ comments

Reviewer #1: This study was designed to investigate the antimicrobial activity of NAC against S. enterica and to verify the potential of NAC as a promising antibiotic for treating S. enterica in combination with other antibiotics. The overall design of the paper is simple and reasonable. Comments: 

(1) The basis for selecting Salmonella ATCC 14028 needs to be supplemented. 

Response: In line 159-163, the discussion section, we explained that we are interested in studying S. enterica due to the multiple outbreaks and food posisoning associated with Salmonella along with the rapid rate of developing antibiotic resistance . We conducted similar research on different Gram negative standard stains (E.coli, Pseudomonas, Shigella) and clinical isolates but it’s still not published. 

(2) The method for determining MIC needs to be supplemented, and the reason for not considering MBC as an indicator should be explained.

Response: Two folds microdilution technique was used as described in line 210-215.

Actually, the concentration mentioned in the manuscript is the MBC as we noticed inhibition of the growth in the well containing (3.125 mg ml-1 of NAC) and a visible growth was shown in the well containing (1.56 mg ml-1 of NAC). Then we tested concentration between the two values mentioned above (3.1, 3, 2.8. 2.5, 2.3 to 1.5) 

We found that at concentration of 3 mg ml-1of NAC showed a full inhibition of S. enterica growth

(3) Figure 2 can be further optimized for display, such as using solid or hollow symbols, solid or dashed lines, etc

Response: Figure 2 is edited.

(4) Is there no color version of Figure 4?

Response: unfortunately, no. Plates and media are transparent; bacteria give a white motility zone, that’s why pictures were captured on a dark background for a better contrast to illustrate the motility zone.

Reviewer #2: The present study tries to establish NAC as an adjuvant which can enhance the effect of antibiotics in treatment of Salmonella Enterica infection. The authors derive the minimal inhibitory concentration of various antibiotics in presence and absence of NAC and show that it can be a beneficial addition to the antibiotic pool. They also show that NAC treatment leads to reduced virulence of S. enterica. Though the study demonstrates the positive role of NAC in combatting S. enterica infection, there are major limitations in the study that need to be addressed.

Major queries:

1) The concentration of NAC used in the study is strikingly different from what is reported in one of the previous studies. Ondrej Chlumsky et.al., 2021 (Evaluation of the Antimicrobial Efficacy of N-Acetyl-l-Cysteine, Rhamnolipids, and Usnic Acid—Novel Approaches to Fight Food-Borne Pathogens) showed MIC of NAC to be 12500 μg/mL which is strikingly different from 3 mg/mL used in the present study. Could the authors justify the effect seen with a lower concentration of NAC?

Response:

1) We think the difference between our MIC values could be due to the starting bacterial load, in our paper NAC was challenged against bacterial concentration of 1x105 CFU ml-1 While Chlumsky et.al., 2021 used bacterial concentration of ( OD =)0.08-0.1 that’s approximately 1x107 CFU ml-1

2) Different strains give different MIC values; we can list multiple papers using NAC against different Gram-negative bacteria. In the above-mentioned paper for Ondrej Chlumsky et.al., 2021 in the second paragraph in the discussion part they said: 

The efficacy of NAC has been earlier mentioned for the inhibition of biofilm formation, disruption of preformed biofilms (both initial and mature), and reduction in bacterial viability in biofilms of clinical pathogens, such as Haemophilus influenza, Pseudomonas aeruginosa, and Streptococcus pneumoniae [15,16,19]. Interestingly, these studies demonstrated lower MIC effectiveness against bacterial growth (500–2500 µg/mL) and needed for biofilm reduction (2500–10,000 µg/mL) than MIC evaluated in our study (3130–12,500 µg/mL for bacterial growth and 12,500–100,000 µg/mL for biofilm reduction). 

Moreover we can mention multiple other studies that give vary range of MIC for NAC For Example:

https://www.ncbi.nlm.nih.gov/pmc/articles/PMC10376233/

MIC for NAC is 4000 μg/mL

https://www.ncbi.nlm.nih.gov/pmc/articles/PMC3251652/

MIC for NAC is 8 mg/ml.

2) The authors show that administration of NAC leads to decreased expression of virulence genes: hilA and fliC. It is important to understand what this would mean physiologically. Does the reduced virulence lead to less severe symptoms of S. enterica infection? Also, it is recommended that the authors show the expression of other important virulence genes such as invA (assists in the host invasion) and iroB (involved in the formation and transfer of enterobactin).

Response:

In our previous work we studied the coordinated regulation of virulence in S. enterica, motility and invasion are tightly coupled. fliC is required for the efficient expression and assembly of FliC into the growing flagellar structure. Noninvasive Salmonella are not pathogenic, invasion is part of the pathogenesis. The hilA gene encodes an OmpR/ToxR family transcriptional regulator that activates the expression of invasion genes, expression of hilA activates the expression of T3-SSS genes including invA. If FliC or HilA expression is affected/ reduced it means the entire infection is aborted. 

3) The results show that the enhanced strains show a lowered MIC for NAC. How would the authors explain this? And why do the virulence genes in NAC resistant strains respond to NAC treatment?

Response: 

In Figure 2, the evolved strains could tolerate a sub-MIC concentration of NAC, grow to a higher OD compare to the wild type but we were curious to determine what is the change in MIC value for the evolved strains, we found that the MIC values are lower than the wild type may be multiple systems are affected and this needs further investigations like RNA sequencing to check exactly what are the genes involved.

May be NAC and other FDA approve substances behave differently than conventional antibiotic, and exposure to sub-MIC is a better option to weaken the pathogen and its virulence factors. 

Recent papers (https://www.ncbi.nlm.nih.gov/pmc/articles/PMC8721600/) reported similar findings with treating Pseudomonas with sub MIC concentrations of different antibiotics, MIC post exposure for 10 cycles either remain constsnt, decreased or increases.

4) Changes in the WGS profile with NAC treatment would not give a substantial explanation to the changes in virulence. An explanation about the probable mechanism of NAC mediated effects on S. enterica needs to be provided in the discussion section. Also, WGS data need to be validated.

Response: 

WGS showed SNPS in multiple genes some of them are constant in all evolved strains and others are unique per strain. We agree that WGS didn’t give a full explanation but at least gave a picture for the possible genes involved in the interacting pathway of NAC and Salmonella. 

An extra paragraph is added to the discussion part (line 197 -201)

Minor queries:

1) The authors use hilA and fliC promoters driven GFP expression as a readout of virulence. However, GFP expressions are not shown in the manuscript. In addition, flow cytometry plots must be shown in the figure along with the quantification graph.

Response: Flow cytometry fplots are added as supplementary documents (S1 Fig, S2 Fig)

2) In the abstract, NAC is referred to as an antibiotic and as an adjuvant in another section. It would be better to maintain a uniform terminology. Also, referring to an antioxidant as an antibiotic might need to be reconsidered.

Response: edited

3) In the discussion section, line 148, it is stated that NAC increases oxidative stress. This misleading statement needs to be corrected.

Response: edited

---

## [Decision Letter · Decision Letter 1]

25 Oct 2024

N-acetyl cysteine exhibits antimicrobial and anti-virulence activity against Salmonella enterica

PONE-D-24-31564R1

Dear Dr. Aboelnaga,

We’re pleased to inform you that your manuscript has been judged scientifically suitable for publication and will be formally accepted for publication once it meets all outstanding technical requirements.

Kind regards,

Anujith Kumar

Academic Editor

PLOS ONE

Additional Editor Comments (optional):

Reviewers' comments:

Reviewer's Responses to Questions

**Comments to the Author**

1. If the authors have adequately addressed your comments raised in a previous round of review and you feel that this manuscript is now acceptable for publication, you may indicate that here to bypass the “Comments to the Author” section, enter your conflict of interest statement in the “Confidential to Editor” section, and submit your "Accept" recommendation.

Reviewer #1: All comments have been addressed

Reviewer #2: All comments have been addressed

2. Is the manuscript technically sound, and do the data support the conclusions?

Reviewer #1: Yes

Reviewer #2: Yes

3. Has the statistical analysis been performed appropriately and rigorously? 

Reviewer #1: Yes

Reviewer #2: Yes

4. Have the authors made all data underlying the findings in their manuscript fully available?

Reviewer #1: Yes

Reviewer #2: Yes

5. Is the manuscript presented in an intelligible fashion and written in standard English?

Reviewer #1: Yes

Reviewer #2: Yes

6. Review Comments to the Author

Reviewer #1: The author has made corresponding revisions, and the suggestions mentioned have been incorporated into the revised manuscript. There are no further comments, and I recommend its acceptance for publication.

Reviewer #2: The authors have provided a satisfactory rebuttal to the concerns raised. The manuscript has been edited wherever necessary. The manuscript can now be accepted for publication in Plos one.

7. PLOS authors have the option to publish the peer review history of their article (what does this mean?). If published, this will include your full peer review and any attached files.

Reviewer #1: **Yes: **Qingli Dong

Reviewer #2: No

---

## [Editor Report · Acceptance letter]

6 Nov 2024

PONE-D-24-31564R1 

PLOS ONE

Dear Dr. Hamed, 

I'm pleased to inform you that your manuscript has been deemed suitable for publication in PLOS ONE. Congratulations! Your manuscript is now being handed over to our production team.

Kind regards, 

on behalf of

Dr. Anujith Kumar 

Academic Editor

PLOS ONE